# On the Design and Lubrication of Water-Lubricated, Rubber, Cutlass Bearings Operating in the Soft EHL Regime

Edward H. Smith 

Jost Institute for Tribotechnology, University of Central Lancashire, Preston PR1 2HE, UK; ehsmith@uclan.ac.uk

**Abstract:** All propeller-driven ships employ a drive shaft supported by journal bearings. To avoid water pollution, these bearings are generally lubricated by the surrounding water, removing the need for a rear seal. Such bearings, commonly referred to as Cutlass bearings, usually have an inner grooved nitrile rubber lining. The grooves (called flutes) allow debris to be flushed out and the bearing surface to be cooled. The remaining area is divided into a number of load-carrying areas called staves. At present, no rigorous design guide exists for these bearings. This paper presents a methodology to predict the minimum film thickness between the journal and the most heavily-loaded stave, an approach not hitherto reported in the literature. The method includes a new, 3D, finite element (FE) approach for soft elasto-hydrodynamic (EHL) predictive modelling of generated pressures in cutlass bearings. Model predictions compare favourably with experimental data. It is shown that the modulus of elasticity of the rubber has no influence on the minimum film thickness. An equation relating dimensionless film thickness to dimensionless load, clearance ratio and numbers of staves is presented. For a nominally circular bearing, increasing the clearance ratio or increasing the numbers of staves reduces load-carrying capacity. It is shown that distortion due to loading can increase load-carrying capacity.

**Keywords:** cutlass bearing; rubber bearing; water lubrication; soft EHL

## 1. Introduction

Rubber-lined journal bearings are widely used to support the propeller shafts of boats and ships, and also the shafts of pumps. The bearings consist of nitrile staves separated by flutes, mounted in a metal or composite housing, as illustrated in Figure 1. In marine applications, the lubricant is usually water rather than oil, thus preventing the egress of oil into the surrounding water. Since the dynamic viscosity of water is so much less than that of typical mineral oils, these bearings operate generally at very low film thicknesses, and sometimes in the mixed lubrication regime. In order to carry reasonable loads, therefore, they are designed with a high length/diameter ratio (typically between two and four). The flutes supply water into the bearing for both lubrication and cooling. If these bearings are correctly aligned, they can operate satisfactorily for many years without any maintenance, suggesting that some form of full fluid-film lubrication pertains. Cabrera et al. [1] suggest that the first water-lubricated rubber bearings were deployed by an American mining engineer called Sherwood. He patented his design and the bearings eventually became known as 'cutlass' bearings, because Sherwood's design was less abrasive than a metal bearing, i.e., it 'cut less'. Cutlass bearings vary in geometry considerably; they can range from 20 to 250 mm in diameter, with anything from 6 to 18 flutes. Radial clearances tend to be much larger than in standard journal bearings. Some manufacturers machine the bores of the bearings to higher accuracies than others. Over the last 20 years, there has been a steadily increasing interest in understanding how cutlass bearings work. Cabrera et al. [1]

undertook pioneering research to investigate the pressures generated in a cutlass bearing, and also performed some soft elasto-hydrodynamic (EHL) analysis aimed at trying to compare experiment with theory. Their eight-flute bearing was 50 mm diameter with a length of 100 mm and a clearance of 0.1 mm (it was not specified if this was radial or diametral). The rubber had an International Rubber Hardness Degrees (IRHD) value of 67 which translates to a Young's modulus of between 4 and 9 MPa (the correlation between the two properties is not a precise one). The surface speed of the journal was maintained at 2 ms$^{-1}$ with loads up to 922 N being applied. Pressures were measured with a transducer mounted in the rotating shaft. It was found that the three staves near the zone of minimum film thickness carried the majority of the load, with maximum pressures of around 37,000 Pa and sub-ambient pressures of almost 20,000 Pa being observed. Normally, any sub-ambient pressures generated in steadily-loaded, oil-lubricated contacts are negligible in relation to their super-ambient counterparts. However, in water-lubricated contacts, the sub-ambient and super-ambient pressures can be of the same order, and this can have an impact on load-carrying capacity. Deformations of the staves produced a saddle-shaped pressure distribution with twin peaks. Numerical difficulties prevented predictions at eccentricity ratios greater than 1.2, and comparisons between theoretical and experimental pressure distributions were not presented. Litwin [2] measured pressures in a 100 mm diameter bearing of length 200 mm and diametral clearance of between 0.4 and 0.5 mm. The bearing was not a conventional one, with the lower half having no flutes, and the upper half being composed of four staves. No information was provided on the hardness of the rubber. Pressures in the lower half were measured with eight transducers mounted in the bearing and values of just over 2 MPa were recorded at a rotational speed of 420 rpm. No sub-ambient pressures were recorded. Zhou et al. [3] studied a 14-flute, 235 mm diameter, 940 mm long bearing with a radial clearance of 0.485 mm. The elastic modulus of the rubber was 13 MPa, with the Shore A hardness quoted as 83. The pressures in the water were measured by a shaft-mounted transducer. At a rotational speed of 225 rpm and a load of 32.7 kN, pressures of almost 140,000 Pa were recorded. Some staves exhibited twin peaks in the pressure distribution across them as observed by Cabrera et al. [1]. The journal centre was found to follow an approximately circular locus of around 0.2 mm radius, representing a variation in eccentricity ratio from 0.25 to 0.95. Thus, the pressures generated do not represent those obtained in a steadily loaded arrangement. Wang et al. [4] examined the performance of water-lubricated, 8-fluted, rubber bearings of 63.7 mm diameter and radial clearances of 0.085, 0.14, and 0.17 mm. The bearings were 100 mm long and rotational speeds of up to 1800 rpm were employed. A finite-difference EHL solution of the Reynolds equation was undertaken, but no pressure distributions were presented, and pressure was not measured in the experiments. Other theoretical EHL studies have been undertaken, for example, [5–11], but experimental measurements of fluid film pressure have not been reported. It is important that pressure measurements are made so that the understanding of the lubrication of these bearings can be enhanced. Because of the geometry employed in the drive-line of a propeller-driven ship, the drive shaft can exhibit considerable vibration, and also mis-alignment can be a problem in some circumstances. As a consequence, there has been a considerable amount of research looking at the vibration characteristics of rubber or composite stern-tube bearings [12–20] along with the effects of misalignment [21].

In order to improve the design of rubber cutlass bearings, it is clearly necessary to develop a theoretical model of their performance that is supported by experimental data. This paper presents such a model and compares it with the measurements presented by Cabrera et al. [1] and then examines the theoretical influence of the number of staves, the clearance ratio and the non-circularity of the bearing. A new EHL equation for minimum film thickness is developed.

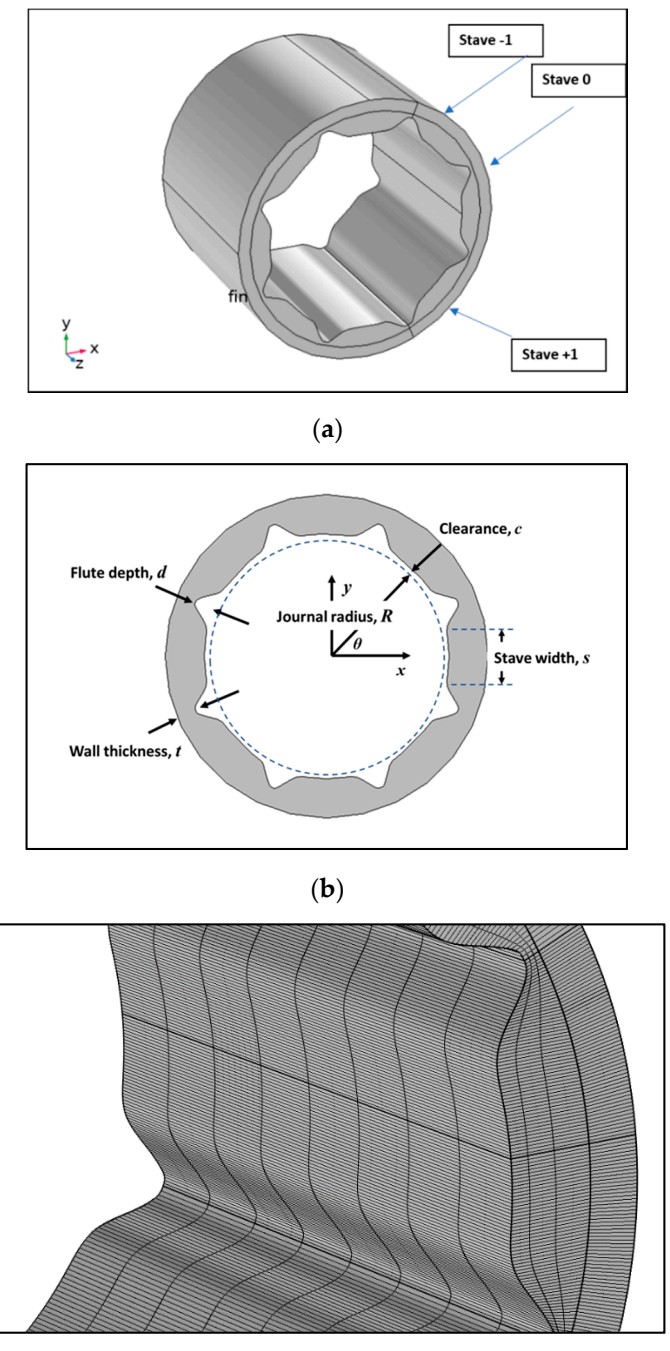

**Figure 1.** Model of the bearing analysed. (**a**) The finite element (FE) model. (**b**) Key geometric variables. (**c**) Mesh geometry at Stave 0.

## 2. Materials and Methods

The pressure in the bearing is governed by the two-dimensional Reynolds equation:

$$\frac{1}{R^2}\frac{\partial}{\partial\theta}\left(\frac{h^3}{12\eta}\frac{\partial p}{\partial\theta}\right) + \frac{\partial}{\partial z}\left(\frac{h^3}{12\eta}\frac{\partial p}{\partial z}\right) = \frac{U}{2R}\frac{dh}{d\theta} \tag{1}$$

where $R$ is the journal radius, $\theta$ and $z$ are the circumferential and axial coordinates, respectively, $h$ is the clearance between the journal and the bearing surface, $U$ is the surface speed of the journal, $\eta$ is the fluid viscosity (assumed to be Newtonian), and $p$ is the gauge pressure in the fluid.

The clearance, $h$ is the sum of three components:

a.　$h_r$—which is the film shape pertaining between an eccentric journal and a circular, or elliptical, bearing

b.　$d$, the additional clearance created by the flutes, and

c.　the radial deformation, $\delta$, of the bearing surface.

The shaft is assumed to be perfectly aligned with the bearing, and thus $h$ is invariant with $y$. The deformation of the rubber bearing is governed by the standard equations of linear elasticity theory [22], namely:

$$\frac{\partial \sigma_x}{\partial x} + \frac{\partial \tau_{xy}}{\partial y} + \frac{\partial \tau_{xz}}{\partial z} = 0; \quad \frac{\partial \tau_{xy}}{\partial x} + \frac{\partial \sigma_y}{\partial y} + \frac{\partial \tau_{yz}}{\partial z} = 0; \quad \frac{\partial \tau_{xz}}{\partial x} + \frac{\partial \tau_{yz}}{\partial z} + \frac{\partial \sigma_z}{\partial z} = 0$$

$$(1+v)\nabla^2 \sigma_x + \frac{\partial^2 \Theta}{\partial x^2} = 0; \quad (1+v)\nabla^2 \sigma_y + \frac{\partial^2 \Theta}{\partial y^2} = 0; \quad (1+v)\nabla^2 \sigma_z + \frac{\partial^2 \Theta}{\partial z^2} = 0$$

$$(1+v)\nabla^2 \tau_{yz} + \frac{\partial^2 \Theta}{\partial y \partial z} = 0; \quad (1+v)\nabla^2 \tau_{xz} + \frac{\partial^2 \Theta}{\partial x \partial z} = 0; \quad (1+v)\nabla^2 \tau_{xy} + \frac{\partial^2 \Theta}{\partial x \partial y} = 0$$

where

$$\Theta = \sigma_x + \sigma_y + \sigma_z$$

and $v$ is the Poisson's ratio.

The EHL problem is solved using finite element (FE) code in COMSOL, with the fluid and elasticity equations fully coupled, and isothermal conditions imposed. An 8-fluted bearing is illustrated in Figure 1. Only half the bearing is analysed because it is assumed that symmetry pertains. A typical mesh for the rubber material is illustrated in Figure 1c. The Reynolds equation is solved using the thin-film flow shell module in COMSOL. The journal is positioned eccentrically inside the bearing, with the minimum film thickness occurring nominally at the centre of stave number 1. The direction of rotation is clockwise. The dimensions of the bearing and journal are taken to be those of the geometry studied by Cabrera et al. [1]. These are presented in Table 1. As noted earlier, Cabrera et al. did not provide the Young's modulus information in their paper, only the rubber hardness number, so it is assumed that the modulus fell within the range 7 to 9 MPa. The bearing is rigidly constrained along its entire outer surface, with symmetry imposed along the centre-line, and the inlet end left free. Zero boundary pressure is applied at the end of the bearing, with symmetry imposed along the centre-line.

**Table 1.** Design data for the bearing.

| Parameter | Value | Unit |
|---|---|---|
| Journal radius, $R$ | 25 | mm |
| Radial clearance, $c$ | 0.05, 0.1, 0.15 | mm |
| Bearing length, $L$ | 100 | mm |
| Wall thickness, $t$ | 4 | mm |
| Flute depth, $d$ | 4 | mm |
| Stave width, $s$ | 11 (approx.) | mm |
| Speed | 765 | rpm |
| Viscosity | 0.0008 | Pa.s |
| Density | 1000 | Kg m$^{-3}$ |
| Young's moduli | 7, 9, 14 | MPa |

No cavitation criterion was applied because experimental evidence [7] suggests that sub-ambient pressures can exist in these bearings and the presence of the flutes is likely to limit the level to which these can extend. Non-circularity was introduced by considering the bearing to have an elliptic shape, with the major axis lying along the line of centres of the journal and bearing, as shown in Figure 2. In this figure, $a$ and $b$ are the major and minor axes of the ellipse, $e$ is the eccentricity of the journal centre, $R$ is the journal radius, $d_1$ and $d_2$ are the distances from the foci to a point on the ellipse, and $f$ is the displacement of the foci from the origin.

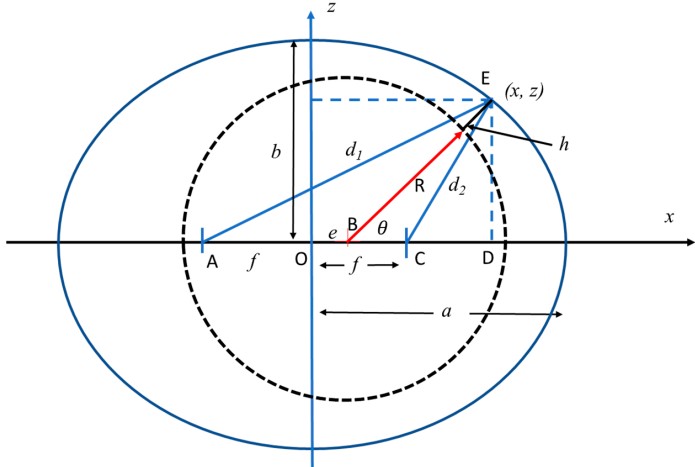

**Figure 2.** Geometry of an elliptic journal bearing.

It is demonstrated in the Appendix A, that the gap, *h*, between the journal and bearing is given by:

$$h \approx \frac{-e\cos\theta + a\sqrt{\frac{a^2}{b^2}\sin^2\theta + \cos^2\theta}}{\left[\cos^2\theta + \frac{a^2}{b^2}\sin^2\theta\right]} - R, \quad when \ e \ll a \tag{2}$$

When the major and minor axes are equal, and have the value $R + c$, this reduces to the familiar equation for a circular bearing, namely:

$$h = c - e\cos\theta \tag{3}$$

where *c* is the radial clearance between the journal and the bearing. (Note, the negative sign in equation 3 exists because $\theta = 0$ refers to the point of nominal minimum film thickness).

## 3. Results

### 3.1. Validation

Illustrated in Figure 3 is a comparison between pressure distributions predicted by the model and one measured by Cabrera et al. [1]. Stave 0 is in the centre, with Staves −1 and +1 being to the left and right of this stave, respectively. Motion is from left to right.

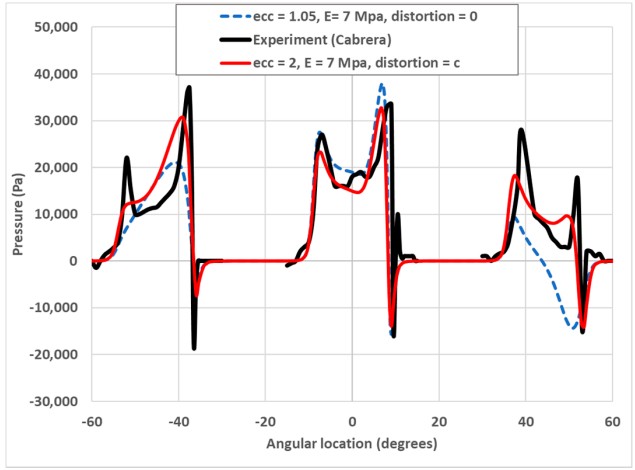

**Figure 3.** Comparison of predicted and measured pressure distributions.

The dotted line shows predictions when the bearing is not distorted. Excellent agreement is achieved on Stave 0, where the 'saddle' shaped pressure distribution is clearly seen. However, agreement is not good on the other two staves, where much higher pressures are measured than those predicted. This suggests that the film thicknesses are lower on these staves than first thought. To test this hypothesis, the bearing was assumed to be an ellipse, with the major axis equal to $R + 2c$, and the minor axis remaining at a value of $R + c$. (Such a distortion could occur due to the applied load). The resulting pressure distribution is plotted in Figure 3. The eccentricity ratio, $\varepsilon = e/c$ in this case was much higher than before, and the predicted pressure distributions on Staves −1 and +1 are much closer in shape to the measured values. Note that the pressure distribution on Stave 0 is substantially the same. It is also clear, that the assumption in the model that there is no cavitation is correct. It is noted that the experimental measurement was taken when the load applied to the bearing was said to be 922 N. Given the magnitude of the measured pressures and the areas over which they act, this seems to be an order of magnitude too high. For the distorted bearing, the model predicts the hydrodynamic force on the journal to be 27.7 N. Finally, the film shapes obtaining at a high eccentricity ratio are plotted in Figure 4. The classical EHL film shape can be seen on the central stave, indicating clearly that soft EHL pertains.

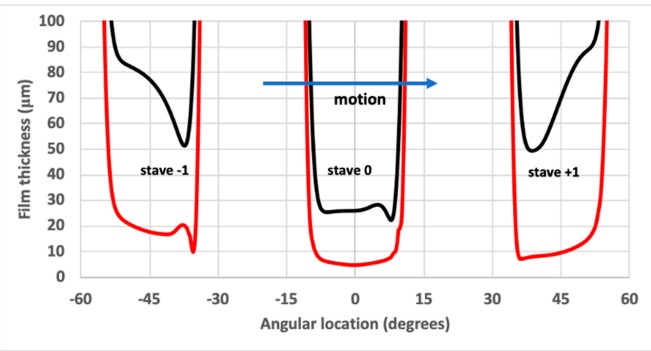

**Figure 4.** Film thicknesses on three staves.

### 3.2. Dimensional Analysis

Generally, it can be assumed that the minimum film thickness in the contact would be given by:

$$h = f(\eta, u, R, c, w^*, E')  \qquad (4)$$

where $\eta$ is the mean viscosity, $u$ is the surface speed of the journal, $R$ is the radius of the journal, $c$ is the radial clearance, $w^*$ is the load per unit axial length, and $E'$ is the reduced elastic modulus. As a first step, the influence of Young's modulus on the load/minimum film thickness relationship was examined and the results are presented in Figure 5. The influence is clearly negligible over the range 7 to 14 MPa considered.

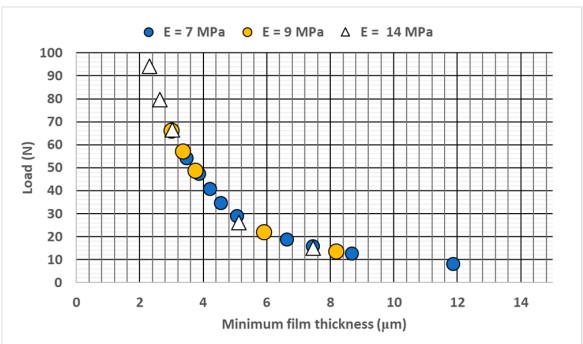

**Figure 5.** Load vs. Minimum film thickness for a range of Young's moduli (circular bearing).

As a consequence, a revised relationship can be written as:

$$h = f(\eta, u, R, c, w^*) \tag{5}$$

where the Young's modulus has been omitted.

Dimensional analysis then reveals that:

$$\frac{h}{c} = f\left(\frac{c}{R}, \frac{w^*}{\eta u}\right) \tag{6}$$

Or

$$H = f(C, W) \tag{7}$$

where, $H = h/c$, $C = c/R$, and $W = w^*/\eta u$.

Extensive runs of the FE model revealed that Equation (7), for a non-distorted bearing with eight staves, can be written as:

$$H = 0.133\ W^{-0.66} C^{-1.25} \tag{8}$$

(Note that in this case, because both surfaces have virtually the same radius, and the bearing is always stationary, the journal radius and velocity are employed in Equation (8), not the equivalent radius and mean entrainment velocity).

Predictions from Equation (8) are compared with those from the FE model in Figure 6. A wide range of clearance ratios, surface speeds and shaft radii were examined, and the fit between the two variables is excellent. When increasing the shaft radius from 25 to 100 mm, the shape of the staves was increased in direct proportion so as to maintain geometric similarity.

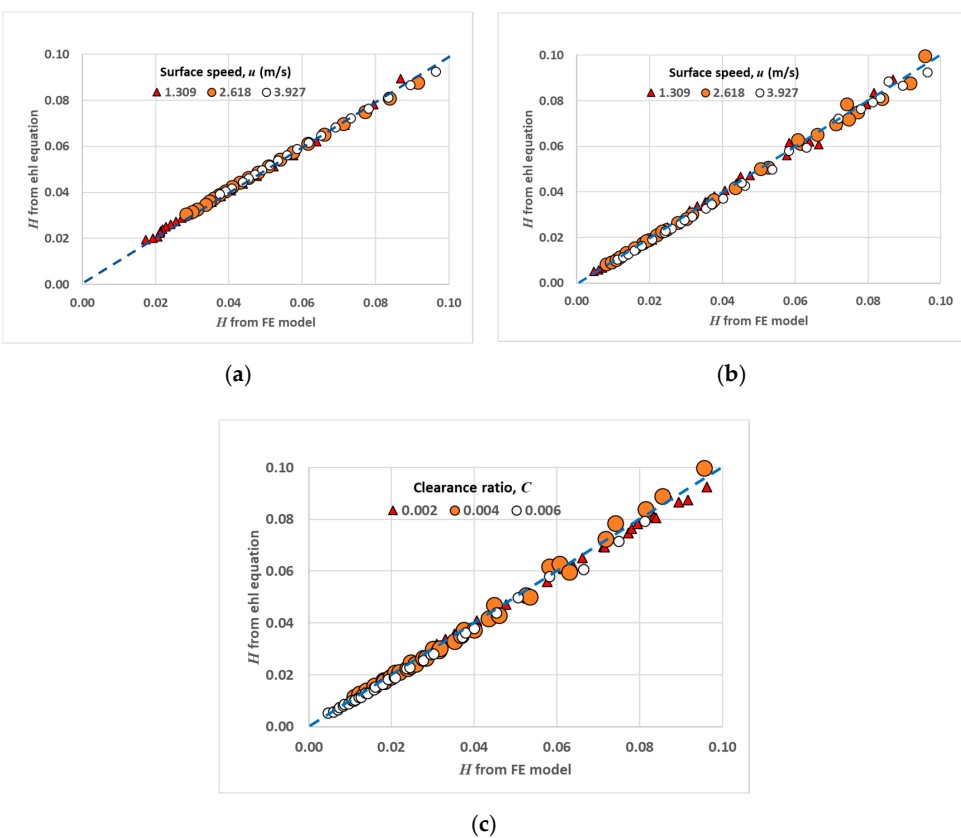

**Figure 6.** Predicted elasto-hydrodynamic (EHL) dimensionless minimum film thickness vs. FE model results. (**a**) R = 100 mm, C = 002, N = 500, 1000, 1500 rpm. (**b**) R = 25 mm, C = 0.002, N = 500, 1000, 1500 rpm. (**c**) R = 25 mm, C = 0.002, 0.004, 0.006, N = 500, 1000, 1500 rpm.

### 3.3. The Effects of Design Parameters

#### 3.3.1. Distortion

The materials and manufacturing processes of cutlass bearings produce components that exhibit some out-of-roundness. To study this effect, the bearing was made elliptical in two ways. Firstly, by increasing the length of the major axis, $a$, (see Figure 2) whilst keeping the minor axis length constant at $(R + c)$, and secondly by increasing the length, $b$, of the other axis whilst keeping $a = (R + c)$.

The effects of distortion on load / minimum film thickness characteristics are shown in Figure 7. The upper curve represents a bearing where the major axis is increased from $a = (R + c)$ to $a = (R + 3c)$, whereas the middle curve is for $a = (R + 2c)$, and the lower curve shows data for a perfectly circular arrangement. Small increases in the size of the major axis can produce large percentage increases in load carrying capacity at the lowest film thickness. At the highest dimensionless loads, the eccentricity ratios were 2.7, 3.7 and 4.5 (with corresponding minimum film thicknesses of 0.98, 0.93 and 0.96 μm) when the distortions were zero, $c$ and $2c$, respectively. Convergence of the analyses became a problem at film thicknesses lower than these values. Cabrera et al. [1] indicated that eccentricity ratios were generally between three and four, suggesting again that their bearing was not perfectly circular.

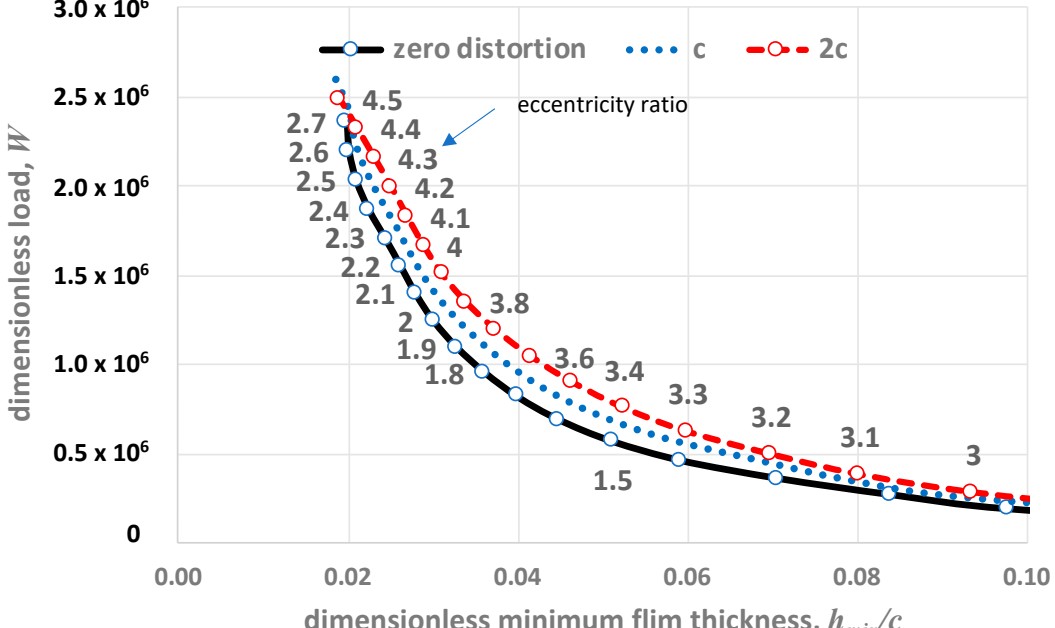

**Figure 7.** The effects of distortion of the major axis, by amounts c and 2c.

Distortion of the other axis of the ellipse is considered in Figure 8. Here the minor axis was increased from $b = (R + c)$ to $b = (R + 3c)$. It is evident that small increases in the size of the minor axis can induce large percentage decreases in load carrying capacity.

It is apparent, therefore, that manufacturing tolerances require precise control if these bearings are to operate most effectively.

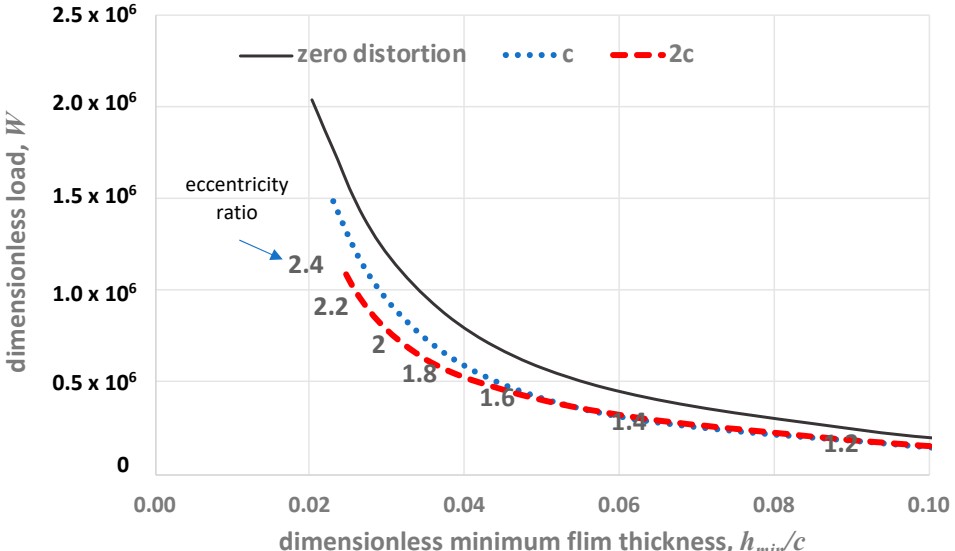

**Figure 8.** The effects of distortion of the minor axis.

### 3.3.2. Number of Staves

Manufacturers provide bearings with a wide number of staves, typically 8 to 14. Figure 9 plots dimensionless minimum film thickness against dimensionless load for a range of staves installed in the bearing discussed earlier. Greater minimum film thicknesses are generated with lower numbers of staves. At the lower operating film thicknesses, it is clearly a considerable advantage to have the lowest number of staves in order to maximise load carrying capacity. Of course, there may be other reasons, such as cooling requirements in the case of mixed or boundary lubrication, where a larger number of staves would be an advantage. The chart also carries predictions (labelled "ehl eq") from a modified version of Equation (8), that is to say:

$$H = 0.732 N^{-0.823} \, W^{-0.66} C^{-1.25} \tag{9}$$

where, $N$ is the number of staves.

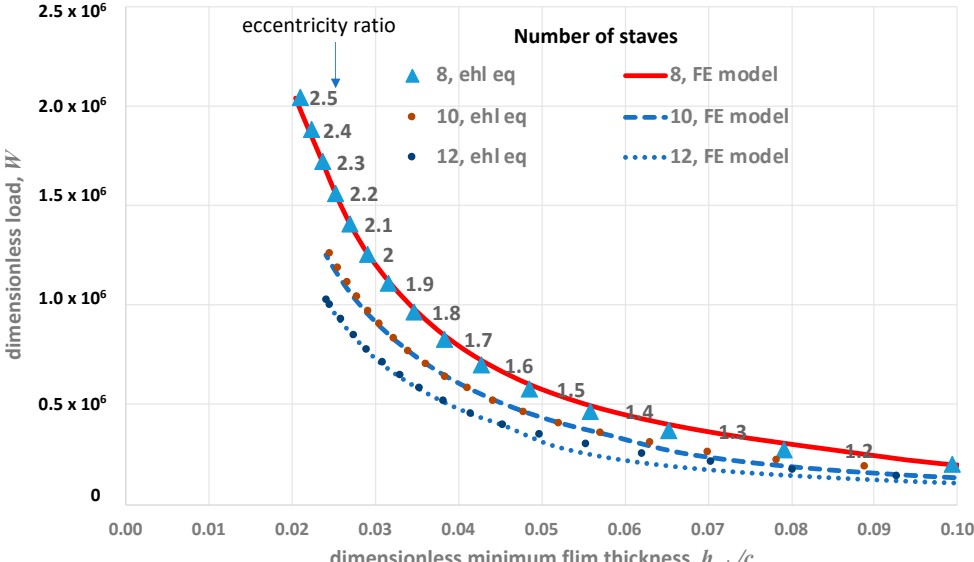

**Figure 9.** The effects of number of staves on bearing performance.

This equation proved to be unreliable for bearings with 14 staves when the non-dimensional film thicknesses were less than 0.04, and for this reason, the predictions for 14 staves is not presented in the figure. This is most likely due to the load support being spread over more staves and the resulting EHL conditions being different on each of them.

### 3.3.3. Clearance Ratio

It has been shown in Equation (8) that the dimensionless film thickness is proportional to the clearance ratio according to:

$$H \propto C^{-1.25}$$

when the dimensionless load parameter is constant.

Thus, for a given bearing load, speed and lubricant,

$$\frac{h}{c} \propto c^{-1.25} \tag{10}$$

or

$$h \propto c^{-0.25} \tag{11}$$

If, for example, a bearing has a radial clearance of 0.1 mm, and it is replaced by a bearing with double the radial clearance, then the minimum film thickness will reduce by approximately 16%. This not an insignificant amount and should be considered when designing cutlass bearings.

### 3.3.4. Stave and Flute Geometry

The EHL equations developed so far have been confined to a specific ratio of journal radius to wall thickness, $t$, and flute depth, $d$, of 25/4. Thus, for example the 25 and 100 mm radii bearings analysed earlier, had wall thickness/flute-depth values of 4 and 16 mm, respectively. Figure 10 demonstrates the effects of changing these proportions in a 10-stave bearing. For reference, the predictions from Equation (8) are included in the chart. The largest change occurs when the original design (flute depth = 16 mm, wall thickness = 16 mm) is changed to 12 mm, 12 mm. The overall effect on performance, however, is not large.

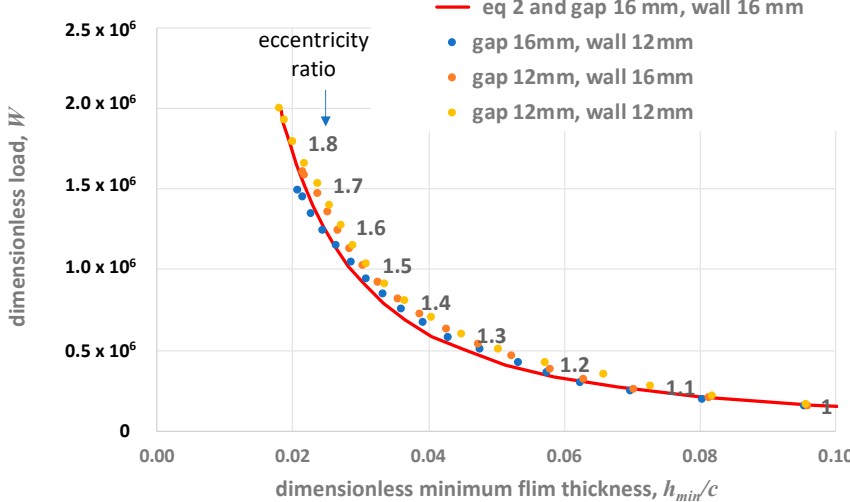

**Figure 10.** The effects of changing wall thickness and flute depth.

## 4. Conclusions

A 3-D, FE model of a cutlass bearing has been developed and its predictions of pressure compare well with experimental data. It is demonstrated that the modulus of elasticity has no influence on

the minimum film thickness in these bearings. An equation relating dimensionless film thickness to dimensionless load and clearance ratio is proposed. This has been developed from a linear regression analysis of film thickness predictions from the FE model using a wide range of input parameters. In particular, it is shown that for a nominally circular bearing with eight staves, the dimensionless minimum film thickness is given by:

$$H = 0.133 \, W^{-0.66} C^{-1.25}$$

This implies that increases in clearance reduce the actual minimum film thicknesses in a given bearing and loading situation. It is also demonstrated that this equation can be modified to account for 8, 10 and 12 staves to become:

$$H = 0.732 N^{-0.823} \, W^{-0.66} C^{-1.25}$$

The negative exponent of the number of staves, *N*, shows that the hydrodynamic load carrying capacity of a nominally circular cutlass bearing reduces as *N* is increased. If the bearing is distorted into an ellipse by increasing slightly its major axis (which lies along the line of journal eccentricity) then the load-carrying capacity can be enhanced. Distortion perpendicular to this direction has the opposite effect. The effects of modest changes in wall thickness and flute depth are seen to have only a small effect on bearing performance. This paper has introduced, for the first time, a rigorous hydrodynamic analysis of water-lubricated, rubber cutlass bearings. It is hoped that this will lay the foundation for further work that will place the design of these bearings on a sound scientific footing.

**Funding:** This work in this paper was funded entirely by the Jost Institute for Tribotechnology, University of Central Lancashire.

**Acknowledgments:** The author would like to thank Mike Cosgrove of Exalto Bearings Ltd. for sharing his extensive and valuable experience. Also, the support given by Duncan Dowson as an inspirational teacher, excellent PhD supervisor, and a kind, generous colleague is gratefully acknowledged.

**Conflicts of Interest:** The author declares no conflict of interest.

**Appendix A**

To consider non-circularity, the bearing, without flutes, was modelled as an ellipse, as shown in Figure A1 with a circular journal at an eccentricity, *e*. The major and minor axes are *a* and *b* respectively. The focal points, A and C, are located at a distance *f* from the centre, O, of the ellipse, where

$$f = \sqrt{(a^2 - b^2)} \ \ and \ d_1 + d_2 = 2a$$

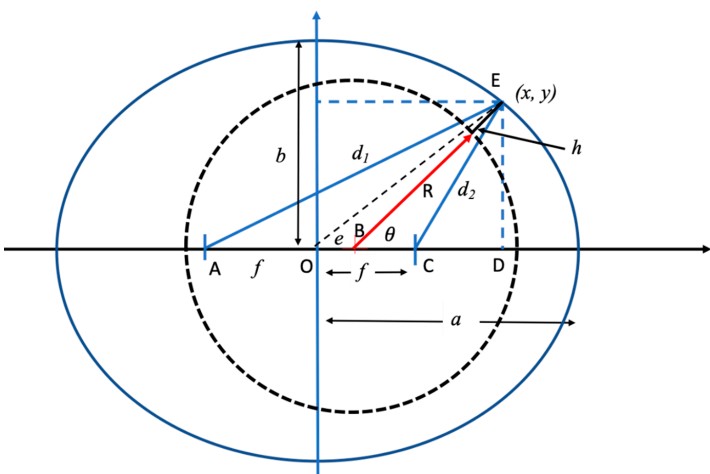

**Figure A1.** A circular shaft in an elliptical bearing.

$$from \ \Delta BED, \quad (R+h)cos\theta = x - e \ \ and \ (R+h)sin\theta = y$$

leading to:

$$x^2 = [(R+h)cos\theta + e]^2 and \ y^2 = [(R+h)sin\theta]^2$$

Using the equation of an ellipse,

$$\frac{x^2}{a^2} + \frac{y^2}{b^2} = 1$$

leads to:

$$h \approx \frac{-ecos\theta + a\sqrt{\frac{a^2}{b^2}sin^2\theta + cos^2\theta}}{\left[cos^2\theta + \frac{a^2}{b^2}sin^2\theta\right]} - R, \qquad when \ e \ll a$$

A similar analysis for distortion of the minor axis can be undertaken.

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
