# Peer review of "On the Design and Lubrication of Water-Lubricated, Rubber, Cutlass Bearings Operating in the Soft EHL Regime"

_lubricants, doi:10.3390/lubricants8070075_

Round 1
Reviewer 1 Report
Minor revision.
Minor english corrections.
Eq. 1 is not in accordance to the referencial shown in figure 1.
Missing information regarding the finite element code:
a) type of elements
b) number of elements
c) time of simulation
d) in-house code or commercial code
Missing information regarding the water properties considered in the simulation:
a) viscosity
b) density
c) rheological model
The viscosity in the reynolds equation is assumed isothermal?
What about the pressure-viscosity dependence?
Author Response
The author would like to thank the reviewer for the constructive and helpful comments. In response to the points raised:
- axes and definition of theta now included in Fig 1
- the use of COMSOL has been noted now, and a typical mesh arrangement included in a new Figure 1(c)
- the viscosity data for water was presented in Table 1. Density has now been added and a note on Newtonian behaviour included.
- The solution was assumed to be isothermal (now noted in text), and the pressures were so low that the pressure-viscosity effect was negligible
The changes requested by you are highlighted in yellow.
Thank you.
Reviewer 2 Report
This manuscript deals with lubrication analysis and design parameters of a cutlass bearing including elastic deformation and elliptical shape. The author shows the clearance shapes, pressure distributions, film thicknesses, and bearing loads and compares some of the results with the published experimental data. This study is of interest and would be useful for engineers. However, I have a few suggestions, which I believe will improve the paper.
The author may wish to describe the basic equations in terms of elastic deformation and force balance, although the reference of Timoshenko and the Reynolds equation are provided.
The reviewer is wondering how the author treats the flow in the flutes and sets the pressure boundary conditions because the Reynolds equation may not be applicable to the region and the pressure distributions are determined by the conditions.
I would recommend to align the x-axes of Figs. 3 and 4.
It would be recommended to add the eccentricity ratios into Figs.5, 7-10.
It would be great if the calculation schema and parameters such as the flow chart, the mesh size, and the convergence criterion were added.
Regarding Fig. 7: I wonder why the left edge solutions of ‘c’ is not shown between those of ‘zero distortion’ and ‘2c’.
In Fig. 9 the author is encouraged to note the reason why the dimensionless load of ‘ehl’ is smaller than that of ‘FE’ for ‘8’ while the dimensionless load of ‘ehl’ is larger than that of ‘FE’ for ‘12’.
Author Response
The author would like to thank the reviewer for the helpful comments. Response to specific points are noted below:
- The elasticity equations have now been included. The boundary conditions are set at the open end (p = 0) and symmetry along the centre-line. It is true that the Reynolds equation may not be applicable in the flutes, but they are so relatively large that the pressure throughout them will be around zero.
- Re aligning the axes. I think the reviewer means using the same labelling. This has now been done.
- Eccentricity ratios added in Figure 7, 8, 9 and 10 to give an indication of what was transpiring.
- I have included a typical mesh in Fig 1(c) and indicated that a commercial code, COMSOL, was employed.
- with regard to Fig 7, I have included the eccentricity ratios for the '2c' case also (it could be too confusing to add in for the 'c' case.
- Finally, with reference to your comments on Figure 9, I think the important issue is that the fit is very good at the higher eccentricity ratios. No fit will, of course, be perfect across a full range of variables.
Round 2
Reviewer 2 Report
The manuscript is revised.